# Investigation of In Vitro and In Vivo Metabolism of α-Amanitin in Rats Using Liquid Chromatography-Quadrupole Time-of-Flight Mass Spectrometric Method

**DOI:** 10.3390/molecules27238612

**Published:** 2022-12-06

**Authors:** Jiyu Lee, Byeong ill Lee, Jangmi Choi, Yuri Park, Seo-Jin Park, Minjae Park, Jeong-Hyeon Lim, Sangsoo Hwang, Jeong-Min Lee, Young G. Shin

**Affiliations:** College of Pharmacy and Institute of Drug Research and Development, Chungnam National University, 99 Daehak-ro, Yuseong-gu, Daejeon 34134, Republic of Korea

**Keywords:** α-amanitin, LC-qTOF-MS, pharmacokinetic, metabolism

## Abstract

The purpose of this study is to investigate the difference of in vitro–in vivo correlation of α-amanitin from clearance perspectives as well as to explore the possibility of extra-hepatic metabolism of α-amanitin. First, a liquid chromatography-quadrupole-time-of-flight-mass spectrometric (LC-qTOF-MS) method for α-amanitin in rat plasma was developed and applied to evaluate the in vitro liver microsomal metabolic stability using rat and human liver microsomes and the pharmacokinetics of α-amanitin in rat. The predicted hepatic clearance of α-amanitin in rat liver microsomes was quite low (5.05 mL/min/kg), whereas its in vivo clearance in rat (14.0 mL/min/kg) was close to the borderline between low and moderate clearance. To find out the difference between in vitro and in vivo metabolism, in vitro and in vivo metabolite identification was also conducted. No significant metabolites were identified from the in vivo rat plasma and the major circulating entity in rat plasma was α-amanitin itself. No reactive metabolites such as GSH-adducts were detected either. A glucuronide metabolite was newly identified from the in vitro liver microsomes samples with a trace level. A semi-mass balance study was also conducted to understand the in vivo elimination pathway of α-amanitin and it showed that most α-amanitin was mainly eliminated in urine as intact which implies some unknown transporters in kidney might play a role in the elimination of α-amanitin in rat in vivo. Further studies with transporters in the kidney would be warranted to figure out the in vivo clearance mechanism of α-amanitin.

## 1. Introduction

An antibody-drug-conjugate (ADC) is one of the interesting classes of targeted therapy and many pharmaceutical companies are working on ADC development [1]. Currently, 11 ADCs have been approved so far by US Food and Drug Administration (FDA) and more than 230 ADC compounds are being developed in the preclinical/clinical development stage [1]. ADC is composed of antibody, linker, and payload for targeting and killing cancer cells. Payloads are normally cytotoxic agents, and the payloads currently used in popular are auristatins, maytansinoids, calicheamicins, duocarmycins, pyrrolobenzodiazepines (PBD), and α-amanitin [1]. Heidelberg Pharma, a German pharmaceutical company, is actively developing ADC programs using α-amanitin as a payload featuring antibody targeted amanitin conjugate (ATAC) platform [2]. HDP-101, an ADC developed by Heidelberg Pharma, is already in the clinical stage targeting multiple myeloma [2].

α-Amanitin is a bicyclic octapeptide derived from mushrooms of the genus Amanita and has cytotoxicity by inhibiting RNA polymerase Ⅱ (Figure 1) [3,4].

In general, understanding the drug metabolism and pharmacokinetics (DMPK) properties of the ADC’s payload is essential for the ADC development and regulatory approval. The DMPK assessment of payload also helps not only to predict the in vivo behavior of ADC but also to understand the pharmacological and toxicological characteristics of ADC [5,6]. So far, limited studies have been conducted for α-amanitin for in vitro experiments and in vivo pharmacokinetics (PK), however, DMPK studies of α-amanitin in rats have not been fully studied yet.

In this study, a liquid chromatography-quadrupole-time-of-flight-mass spectrometric (LC-qTOF-MS) method for α-amanitin in rat plasma was first developed and applied for PK study of α-amanitin via intravenous (IV) or oral (PO) administration in rats. In vitro metabolic stability study in liver microsomes as well as metabolite identification (MetID) study were also performed to assess whether the metabolic behavior of α-amanitin was different between in vitro and in vivo environments. Finally, a semi-mass balance study in rat was conducted to explore the major elimination route of α-amanitin. To our best knowledge, this is the first paper to explore the in vitro–in vivo correlation of α-amanitin using the PK, in vitro/in vivo MetID and semi-mass balance studies to better understand the DMPK properties of α-amanitin in rats.

## 2. Results and Discussion

### 2.1. Method Development and Qualification

The method for quantification of α-amanitin in rat plasma was developed in this study. The calibration curve consists of duplicate standard (STD) samples at seven levels of concentration and quality control (QC) samples at three levels. The calibration curve was established in the range of 3.05–2220 ng/mL by the quadratic regression (weighted 1/concentration) with a correlation coefficient > 0.99 which is sufficient to cover the PK of α-amanitin. The calibration curve and the chromatogram of the lower limit of quantification (LLOQ) for α-amanitin in rat plasma are shown in Figure 2.

Assay performance was determined by assessing the mean accuracy and precision of QC samples with three levels of concentrations (15.0, 165, and 1820 ng/mL). The intra and inter-run accuracy and precision were evaluated using triplicates and the results are shown in Table 1. The values (%) were all within ±25% of accuracy.

To evaluate the stability of α-amanitin in rat plasma under the actual experimental environments, some stability tests were also conducted. The results of the stability tests are shown in Table 2. The accuracy (%) and precision (% CV) of the stability test met the acceptance criteria of ±25%, identifying α-amanitin was stable under the short-term, long-term, and freeze-thaw conditions.

The extraction recovery of α-amanitin after protein precipitation for sample preparation was 30.7 ± 3.16%. The robust results are shown in Table 3. Although the extraction recovery of α-amanitin in rat plasma was relatively low in this experiment, the accuracy and precision of the calibration curve samples and QC samples were all within the acceptance criteria (Table 1 and Table 2) and therefore the impact of the recovery from this extraction procedure was not significant. Nevertheless, further extraction methods to increase the recovery would be continued for other biological matrices in the future.

### 2.2. In Vitro Metabolic Stability in Liver Microsomes

In vitro stability of α-amanitin at liver microsomal system was evaluated by a LC-qTOF-MS. The remaining % represents the percentage of the sample (α-amanitin) peak area to the internal standard (verapamil) peak area. The remaining % according to incubation time is presented in Table 4. The incubation time to ln (Remaining %) profile is shown in Figure 3.

The in vitro half-life (T_1/2_) and the in vitro intrinsic clearance (CL_int, in vitro_) were calculated by using the slope (*k*) of the graph shown in Figure 3. The calculation formula is shown in Equations (1) and (2) [7].
(1)$$T_{1/2}=\frac{\ln\left(2\right)}{k}$$
(2)$${\mathrm{CL}}_{\mathrm{int},\mathrm{in}\;\mathrm{vitro}}\left(\mathrm{mL}/\min/\mathrm{mg}\right)=k\times \frac{\mathrm{microsomal}\;\mathrm{volume}\;\left(\mathrm{mL}\right)}{\mathrm{microsomal}\;\mathrm{protein}\;\left(\mathrm{mg}\right)}$$

The concentration of liver microsomes (microsomal volume/microsomal protein) used in the experiment was 0.5 mL/mg. The obtained CL_int, in vitro_ was used to calculate in vivo intrinsic clearance (CL_int, in vivo_). The equations for calculating CL_int, in vivo_ is shown at Equation (3) [7]:(3)$${\mathrm{CL}}_{\mathrm{int},\mathrm{in}\;\mathrm{vivo}}\left(\mathrm{mL}/\min/\mathrm{kg}\right)={\mathrm{CL}}_{\mathrm{int},\mathrm{in}\;\mathrm{vitro}}\times \left(\frac{\mathrm{mg}\;\mathrm{microsomal}\;\mathrm{protein}}{g\;\mathrm{liver}\;\mathrm{weight}}\right)\times (\frac{g\;\mathrm{liver}\;\mathrm{weight}}{\mathrm{kg}\;\mathrm{body}\;\mathrm{weight}})$$

The ratio of mg microsomal protein to g liver weight was 44.8 and 48.8 in rat and human, respectively [8]. And the ratio of g liver weight/kg body weight was 40 and 25.7 in rat and human, respectively [8]. The hepatic clearance (CL_h_) was calculated from CL_int_, in vivo in a well-stirred model using the following Equation (4) [8]:(4)$${\mathrm{CL}}_{h}\left(\mathrm{mL}/\min\right)=(\frac{Q_{h}\times {\mathrm{CL}}_{\mathrm{int},\mathrm{in}\;\mathrm{vivo}}}{Q_{h}+{\mathrm{CL}}_{\mathrm{int},\mathrm{in}\;\mathrm{vivo}}})$$

Q_h_ (mL/min/kg), the hepatic blood flow, is known to be 55.2 and 20.7 mL/min/kg in rat and human, respectively [8]. The calculated metabolic stability parameters of α-amanitin in rat and human microsomes are shown in Table 5. As a result, α-amanitin has a low hepatic clearance according to the aforementioned well-stirred model.

### 2.3. Application for PK Study in Rat

The above developed LC-qTOF-MS method was applied to obtain the PK parameters after IV and PO administration in rat. Most of the PK samples were within the qualified calibration curve, except the 2 min-time point samples in the IV group which were slightly above the upper limit of quantification (ULOQ). Since the calibration curve was linear in the range of quantification, the concentrations of these early time point IV samples have been calculated by extrapolation of the calibration curve which is also applicable for discovery study in pharmaceutical industries [9]. The time to concentration profile of α-amanitin after IV administration is shown in Figure 4. The PK parameters were calculated with the non-compartmental analysis (NCA) using WinNonlin^®^ version 8.1.0 (Certara, Princeton, NJ, USA) and the results are summarized in Table 6. The results of the PO administration are not shown, because all PO samples were below the LLOQ, which was likely due to the limited absorption process in the gastrointestinal tracts.

The abbreviations shown are as follows: T_1/2_ (half-life), C_max_ (the observed maximum concentration in plasma), AUC_last_ (area under the concentration–time curve to the last time point), and V_ss_ (volume of distribution at steady state). The PK parameters of α-amanitin in this rat study showed comparable values to those in mice and dogs [10,11].

### 2.4. Semi-Mass Balance Study in Rat

The excretion characteristic of α-amanitin after IV and PO administration is shown in Table 7, and the time to recovery of α-amanitin profile is shown in Figure 5. The recovery represents the percentage of the excreted amount (mg) to the administrated amount (mg).

In the IV group, the total recovery of α-amanitin was 68.9% and most of the administered α-amanitin was excreted in urine within 8 h, while the total recovery of α-amanitin in PO group was only 23.5%, and most of the detected α-amanitin was excreted in feces. The recovered α-amanitin after PO administration in feces would be likely the non-absorbed α-amanitin but the recovery was much less than anticipated. There are a couple of possibilities about the low excretion of α-amanitin after PO administration and one is probably due to the limited collection time for the semi-mass balance study (although we used more than five times the half-life of α-amanitin for the semi-mass balance study). The other likelihood of the low excretion of α-amanitin in feces is its binding to the surface of GI tracts. The latter could be explained if the α-amanitin was quantitated from the GI tracts after homogenization. Another approach would be to administer charcoal to the rats after a semi-mass balance experiment to quantitate the remaining amount of α-amanitin in the GI tracts. Either way, further investigation would be warranted to understand the mass balance characteristics of α-amanitin in rats.

### 2.5. In Vitro and In Vivo MetID for α-Amanitin

Although the predicted hepatic clearance of α-amanitin in rat liver microsomes was quite low (5.05 mL/min/kg), the in vivo clearance from the rat PK study (14.0 mL/min/kg) was relatively higher than anticipated which implies that either extra-hepatic clearance or different metabolism in vivo might play a role. Therefore, in vitro and in vivo MetID were conducted to explore any possible different metabolites between the two systems. The in vitro and in vivo MetID samples were analyzed by a LC-qTOF-MS, and the data were used to elucidate a metabolite of α-amanitin using PeakView^®^ and MetabolitePilot™. The structure of the metabolite of α-amanitin was predicted by MetabolitePilot™. The fragmentation pattern of the predicted metabolite ion from product ion scan was compared to the fragmentation pattern of the parent ion using PeakView^®^.

α-Amanitin had a parent ion of *m/z* 919.36 ([M+H]^+^) with a retention time of 9.03 min. The product ion of *m/z* 919.36 was formed by *m/z* 143.12, 171.11, 259.13, 373.18, 749.26, 806.28, and 901.35. These fragment ions were used to prove that the predicted metabolite was derived from α-amanitin. The fragmentation pattern of *m/z* 919.36 in mass spectrometry is shown in Figure 6.

The predicted metabolite ion was *m/z* 548.20 ([M+2H]^2+^), suggesting that 176 amu was increased through metabolism. This suggests that the α-amanitin may have undergone glucuronidation (+C_6_H_8_O_6_, 176.0321 amu). After performing product ion scan of *m/z* 919.36 (parent) and 548.20 (a glucuronide metabolite), the two fragmentation patterns were compared, and as a result, a glucuronide metabolite was generated only in CLM and HLM, not in MLM, RLM, DLM or in vivo rat plasma. In addition, no other significant metabolites of α-amanitin were identified from the in vivo rat plasma samples and the major circulating metabolic component of α-amanitin was α-amanitin itself. The compared fragmentation patterns of parent ion and metabolite in CLM and HLM are shown in Figure 7.

The fragment ions with *m/z* 171.11, 259.13, and 373.18 were found in non-metabolized form, and the fragment ion with 539.20 ([M+2H]^2+^) was found in metabolized form, increasing 176 amu from *m/z* 901.35. Consequently, the characterization of α-amanitin and its metabolite is shown in Table 8, and the predicted metabolic pathway of α-amanitin in CLM and HLM is shown in Figure 8.

The total ion chromatogram (TIC) of parents and a proposed glucuronide metabolite of α-amanitin is shown in Figure 9.

## 3. Materials and Methods

### 3.1. Reagents and Chemicals

α-Amanitin was purchased from Biosynth Carbosynth (Compton, Berkshire, United Kingdom). Verapamil, glutathione (GSH), and uridine-5′-diphosphoglucuronic acid triammonium salt (UDPGA) were purchased from Sigma-Aldrich (St. Louis, MO, USA). Dimethyl sulfoxide (DMSO), formic acid, and methanol (MeOH) were purchased from Dae-Jung reagents (Siheung, Gyeonggi, Republic of Korea). Blank plasma from male Sprague Dawley (SD) rats treated with heparin as anti-coagulant was purchased from Biomedex (Seoul, Republic of Korea). Acetonitrile (ACN) and distilled water (DW) were purchased from Samchun Chemical (Pyeongtaek, Gyeonggi, Republic of Korea). Mouse, rat, cynomolgus monkey, dog, and human liver microsomes (MLM, RLM, CLM, DLM, and HLM), β-nicotinamide adenine dinucleotide hydrate (NADPH) regenerating system solution A (26 mM NADP^+^, 66 mM glucose-6-phosphate, and 66 mM MgCl_2_ in water), and B (40 U/mL glucose 6-phosphate dehydrogenase in 5 mM sodium citrate) were purchased from Corning (Tewksbury, MA, USA). Other chemicals and reagents were purchased from commercial sources.

### 3.2. Preparation of Stock Solution, Calibration Standard (STD), and Quality Control (QC) Samples

#### 3.2.1. Stock Solution, Sub-Stock Solution, and Working Solutions for STD and QC

A stock solution of α-amanitin was prepared by dissolving powder compound in DMSO to make the concentration to 1 mg/mL and stored at –20 °C. A sub-stock solution was prepared by diluting the stock solution in DMSO to a concentration of 0.1 mg/mL. The sub-stock solution was serially diluted with DMSO to obtain a final concentration of 3.05, 9.14, 27.4, 82.3, 247, 741, and 2220 ng/mL for STD working solutions and 15.0 (low) 165 (medium) and 1820 (high) ng/mL for QC working solutions.

#### 3.2.2. STD and QC Samples for Calibration Curve in Pharmacokinetic (PK) Study

Seven calibration STD of α-amanitin were prepared in duplicate by spiking 8 μL of freshly prepared STD working solutions into 40 μL of blank rat plasma. Three levels of QCs were prepared by spiking 8 μL of freshly prepared QC working solutions into 40 μL of blank rat plasma.

#### 3.2.3. STD and QC Samples for Calibration Curve in a Semi-Mass Balance Study

Seven calibration STD of α-amanitin were prepared in duplicate by spiking 8 μL of freshly prepared STD working solutions into 40 μL of a mixed blank matrix (rat blank urine: rat blank bile: rat blank feces: DW = 1:1:1:1, *v/v/v/v*). Rat blank urine were prepared with 30% ACN in DW in a ratio of 1:1 (*v/v*). Rat blank feces were ground using a blender and mixed with phosphate buffered solution (PBS) and ACN (feces: PBS: ACN = 1:9:5, *w/v/v*), then homogenized by beads. Three levels of QCs were prepared by spiking 8 μL of freshly prepared QC working solutions into 40 μL of the mixed blank matrices.

### 3.3. Sample Preparation

#### 3.3.1. Sample Preparation for PK Study Samples

For the plasma PK sample, 8 μL of make-up DMSO was spiked into 40 μL of rat PK samples to make them in identical matrix condition as STD or QC samples. Then, 200 μL of internal standard solution (20 ng/mL of verapamil in ACN) was spiked into STD, QC, and PK samples and vortexed for 30 s. The samples were centrifuged at 12,000 rpm for 5 min. After centrifugation, 200 μL of the supernatant was transferred into a different 1.5 mL Eppendorf tube (E-tube), and evaporated to dryness under vacuum using rotary evaporator (Eyela CVE-3110 and UT-1000, Tokyo, Japan) for 3 h. Evaporated samples were reconstituted with 60 μL of 50% MeOH with 5 mM ammonium formate, and vortexed for 30 s. The reconstituted samples were centrifuged at 12,000 rpm for 5 min and then 50 μL of the supernatant was transferred to a LC vial for LC-qTOF-MS analysis.

#### 3.3.2. Sample Preparation for Semi-Mass Balance Samples

Urine collected from rats were diluted two-fold with 30% ACN in DW. Feces collected from rats were diluted 15-fold with PBS and ACN (feces: PBS: ACN = 1:9:5, *w/v/v*) and homogenized by beads.

For urine samples, 8 μL of make-up DMSO was spiked into 40 μL of a mixed urine sample (urine sample: blank bile: blank feces: DW = 1:1:1:1, *v/v/v/v*). For bile samples collected from rats, 8 μL of make-up DMSO was spiked into 40 μL of a mixed bile sample (bile sample: blank urine: blank feces: DW = 1:1:1:1, *v/v/v/v*). For feces samples, 8 μL of make-up DMSO was spiked into 40 μL of a mixed feces sample (feces sample: blank urine: blank bile: DW = 1:1:1:1, *v/v/v/v*).

For sample pretreatment, 200 μL of internal standard solution (20 ng/mL of verapamil in ACN) was spiked into STD, QC, and semi-mass balance samples and vortexed for 30 s. The samples were centrifuged at 12,000 rpm for 5 min. After centrifugation, 200 μL of the supernatant was transferred into a different tube, 1.5 mL E-tube, and evaporated using the rotary evaporator for 3 h. Evaporated samples were reconstituted with 60 μL of 50% MeOH with 5 mM ammonium formate and vortexed for 30 s. The reconstituted samples were centrifuged at 12,000 rpm for 5 min, and then 50 μL of supernatant was transferred to a LC-vial for a LC-qTOF-MS analysis.

#### 3.3.3. Sample Preparation for In Vitro Metabolic Stability Test in Liver Microsomes

A metabolic stability test of α-amanitin was performed in rat and human liver microsomes. Cofactors (NADPH regenerating system solution A and B), compound (100 μg/mL of α-amanitin), 0.5 M potassium phosphate buffer with pH 7.4, and DW were mixed and pre-incubated for 5 min at 37 °C. After the pre-incubation, 12 μL of rat or human liver microsome (20 mg/mL each) was added to a 468 μL of pre-incubated cofactor mixture and the mixture was incubated for 0, 15, 30, and 60 min. After the incubation for each time point, 100 μL of the mixture was aliquoted to a 1.5 mL E-tube at each time point and 100 μL of 50% ACN in MeOH which containing 250 ng/mL of verapamil as an internal standard for protein precipitation was added for stopping the reaction. Then the mixture was centrifuged at 10,000 rpm for 5 min. Then 180 μL of the supernatants was collected in a fresh E-tube and evaporated for 3 h. After evaporation, samples were reconstituted with 150 μL of 50% MeOH with 5 mM ammonium formate. After centrifugation at 10,000 rpm for 5 min, 140 μL of reconstituted sample was transferred to a LC-vial.

#### 3.3.4. Sample Preparation for In Vitro/In Vivo MetID

In vitro MetID was conducted in 5 species (mouse, rat, cynomolgus monkey, dog, and human) of liver microsomes. Cofactors (NADPH solution A and B, 5 mM UDPGA, and 0.5 mM GSH), compound (2 mg/mL of α-amanitin), and DW were mixed and pre-incubated for 5 min at 37 °C. After the pre-incubation, 20 μL of liver microsome (20 mg/mL) was added to 380 μL of cofactor mixture and incubated for 0 and 120 min at 37 °C. After the incubation for each time point, 150 μL of the mixture was transferred to 450 μL of 50% ACN in MeOH for stopping the reaction. Then the mixture was centrifuged at 10,000 rpm for 5 min. Then 550 μL of the supernatant was collected in a fresh E-tube and evaporated for 4 h. After evaporation, samples were reconstituted with 60 μL of 50% MeOH with 5 mM ammonium formate. After centrifugation at 10,000 rpm for 5 min, 50 μL of reconstituted sample was transferred to a LC-vial.

For rat in vivo MetID, rat plasma samples collected after IV administration of α-amanitin with the dose of 1 mg/kg were pooled according to the Hamilton pooling method [12]. Then 3 μL of DMSO was added to the 300 μL of pooled rat plasma. A control sample was made by mixing 300 μL of blank rat plasma and 3 μL of 0.5 mg/mL α-amanitin. Eight hundred microliter of 50% ACN in MeOH was added to the pooled rat plasma mixture and a control sample, respectively. Then the mixture was centrifuged at 10,000 rpm for 5 min. Then 900 μL of the supernatants was collected to a fresh E-tube and evaporated for 4 h. After evaporation, samples were reconstituted with 120 μL of 50% MeOH with 5 mM ammonium formate. After centrifugation at 10,000 rpm for 5 min, 100 μL of reconstituted sample was transferred to a LC-vial.

### 3.4. LC-qTOF-MS Conditions

The LC-qTOF-MS method consisted of a chromatographic pump system (Shimadzu CBM-20A/LC-20AD, Shimadzu Corporation, Columbia, MD, USA), an auto-sampler system (Eksigent CTC HTS PAL, LEAP Technologies, Carrboro, NC, USA), and a quadrupole time-of-flight mass spectrometer (TripleTOFTM 5600, Sciex, Foster City, CA, USA) with an ion source (Duospray™, Sciex, Foster City, CA, USA).

A Kinetex XB-C18 analytical column (2.1 mm × 50 mm, 2.6 μm; Phenomenex) was used for bioanalytical sample quantification and a Hydro-RP analytical column (2 mm × 100 mm, 2.5 μm; Phenomenex) was used for MetID. Security Guard Cartridge (4 mm × 2 mm; Phenomenex) was placed on the upstream of the analytical column. The LC mobile phase was DW containing 0.1% formic acid for the mobile phase A and ACN containing 0.1% formic acid for the mobile phase B. The LC gradient is shown in Table 9. The flow rate was 0.4 mL/min and the injection volume was 10 μL.

#### 3.4.1. TOF-MS Method for Quantification of α-Amanitin

For the TOF-MS scan, the scan range was *m/z* 100–1000. For the product ion scan, the parent ion of α-amanitin was *m/z* 919.36 ([M+H]^+^) and a product ion at *m/z* 901.35 was used as a quantitative ion (DP: 10 and CE: 31). The parent ion of verapamil (internal standard) was *m/z* 455.3 ([M+H]^+^), and a product ion at *m/z* 165.1 was used as a quantitative ion (DP: 125 and CE: 30). Gas sources 1 and 2 were set to 50 psi and the curtain gas flow was 30 L/min. The ion spray voltage (ISVF) was 5500 V, and the source temperature was 500 °C.

#### 3.4.2. TOF-MS Method for MetID of α-Amanitin

The high-resolution TOF full scan was used for metabolite profiling. The scan range was *m/z* 50–1300 (DP: 50 and CE: 10). In product ion mode, α-amanitin was set to *m/z* 919.36 (DP: 10 and CE: 31), and the suspected metabolite was set to *m/z* 548.20 (DP: 10 and CE: 10) to determine the metabolite. Gas sources 1 and 2 were set to 50 psi and the curtain gas flow was 35 mL/min. The ISVF was 5500 V and the source temperature was 500 °C. The MetID analysis was done by the Information-Dependent Analysis (IDA) method which includes both a real time multiple mass defect filtering (MDF) function and a dynamic background subtraction function.

### 3.5. Method Qualification in Rat Plasma

The method qualification was performed with “fit-for-purpose” criteria which are suitable for discovery stage research. The qualification contained seven levels of STD and three levels of QC in duplicate. To prove the qualification of the bioanalytical method, intra- and inter-day precision and accuracy were also conducted on three different days based on our in-house criteria. Preliminary stability tests were performed at low, medium, and high QC in rat plasma. The stability test was performed under three different conditions: a short-term, a long-term, and a freeze-thaw condition. The short-term stability test was conducted at room temperature (RT) for 4 h. The long-term stability was conducted at –20 °C for 2 weeks. The freeze-thaw stability was conducted for three freeze and thaw cycles at –20 °C and RT. The extraction recovery of protein precipitation was evaluated by analyzing the difference between extracted Medium-QC (165 ng/mL) samples and post-extraction QC samples. The post-extraction QC sample was prepared by spiking the α-amanitin after protein precipitation.

The precision and accuracy of the acceptance criteria for this qualification runs were within ± 25% which are also similar to the discovery-stage bioanalytical criteria in pharmaceutical industry [13]. All experiments were performed in triplicate or more.

### 3.6. The PK Study in Rats

Animal Male SD rats (320 ± 10 g) were purchased from the Samtako Biokorea co. (Gyeonggi, Republic of Korea) and housed in a group of four units per cage and given standard rodent chow. The rats were fasted overnight with free access to water for 12 h before administration. The formulation was prepared by dissolving α-amanitin in normal saline (NS) and DMSO (4:1, *v/v*).

The rats with femoral artery and femoral vein cannulation were prepared for the IV administration, and the rats with single femoral artery cannulation were prepared for the PO administration. The cannulated SD rats were given 1 mg/kg of α-amanitin by either IV bolus or PO (n = 4, both). After administration of α-amanitin, blood sample was collected through a femoral artery cannula and transferred into a heparin-containing tube at each sampling time point (IV: 2, 5, 15, 30, 60, 120, 240, 480, and 1440 min after drug administration and PO: 5, 15, 30, 60, 120, 240, 480, and 1440 min after drug administration). Then plasma sample was obtained by centrifuging the blood sample at 13,000 rpm for 5 min. The obtained plasma samples were stored at –20 °C until analysis. After analysis, the PK parameter was calculated using WinNonlin^®^.

### 3.7. Semi-Mass Balance Study in Rats

A semi-mass balance study was performed to explore the major elimination route of α-amanitin in rats. A non-radiolabeled α-amanitin was used for the semi-mass balance study which seems acceptable due to low metabolic turn-over of α-amanitin.

For the semi-mass balance study, the housed rats were prepared with a bile duct cannulation and administrated by IV bolus in a lateral tail vein or by PO at 1 mg/kg for each dosing route. Urine and feces were collected using a metabolic cage (Jeungdo Bio & Plant co., Seoul, Republic of Korea), and bile was collected through a bile duct cannula. The sampling time interval were 0–4, 4–8, 8–24, and 24–48 h after administration. The urine, bile, and feces samples were stored at –20 °C until further analysis.

### 3.8. Software

Data acquisition and a LC-qTOF-MS operation were conducted using Analyst^®^ TF Version 1.6 (Sciex, Foster City, CA, USA). α-Amanitin was quantified by MultiQuant^®^ Version 2.1.1 (Sciex, Foster City, CA, USA) using peak integration. The PK parameters of α-amanitin were calculated by WinNonlin^®^ version 8.1.0 (Certara, Princeton, NJ, USA) in a non-compartment analysis. For MetID analysis, PeakView^®^ Version 2.2 (Sciex, Foster City, CA, USA) and MetabolitePilot™ Version 2.0.2 (Sciex, Foster City, CA, USA) were used for the structural elucidation of α-amanitin metabolites. Excel 2016 spreadsheet (Microsoft^®^) was also used to process the statistical analysis of results.

## 4. Conclusions

α-Amanitin is recently acknowledged as one of the most popular payloads of ADC and many researchers and pharmaceutical companies are interested in the application of α-amanitin to their ADC research. However, the ADME/PK studies of α-amanitin in various preclinical species are still in premature stages so far. To understand the ADME/PK properties of α-amanitin in rats better, we have explored several in vitro and in vivo studies including bioanalytical method qualification, in vitro liver microsomal stability, in vivo/in vitro MetID, in vivo semi-mass balance study, and in vivo IV/PO PK study. A LC-qTOF-MS method for quantifying α-amanitin in rat plasma was well developed over the calibration range from 3.05 to 2220 ng/mL for quadratic regression with a correlation coefficient > 0.99.

The in vitro metabolic stability test incubated with UDPGA and GSH suggested that α-amanitin rarely underwent liver microsomal metabolism and the predicted clearance was quite low. Moreover, no reactive metabolites with GSH adducts were detected. However, a glucuronide metabolite was found in CLM and HLM as a minor metabolite from in vitro MetID study. To our best knowledge, this is the first report of the glucuronide metabolite of α-amanitin.

The PK study of α-amanitin in rat demonstrated that α-amanitin had a clearance close to 30% of hepatic blood flow and very low bioavailability, which was different from what we anticipated from the in vitro liver microsomal stability. To investigate the possibility of extra-hepatic metabolism, in vivo MetID was also conducted to compare with in vitro MetID results. No difference was observed between the two matrices, and the major circulating entity in in vivo rat plasma sample was still α-amanitin.

Therefore, semi-mass balance study was conducted to explore the contribution of in vivo α-amanitin clearance from the elimination perspective. The semi-mass balance study showed that most α-amanitin was eliminated very quickly in urine within 8 h after IV administration which suggests that some renal transporters in the kidney might actively play a role in terms of in vivo clearance of α-amanitin. Recently US FDA also indicated the importance of several transporters such as OATP1B1, OAT, OCT, etc., in terms of drug–drug interaction and since the renal excretion is the main elimination route of α-amanitin, the evaluation of transporters for α-amanitin would be quite necessary for its ADC drug development in the future [14]. Further studies will be warranted to understand the roles of transporters in renal clearance of α-amanitin as well as the in vivo MetID of α-amanitin in urine.

## Figures and Tables

**Figure 1 molecules-27-08612-f001:**
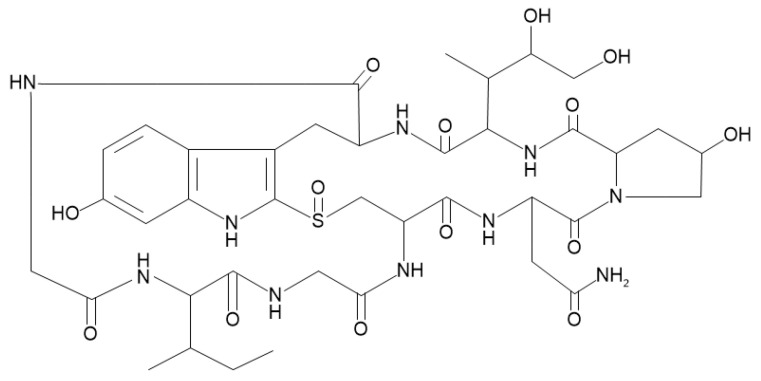
Structure of α-amanitin.

**Figure 2 molecules-27-08612-f002:**
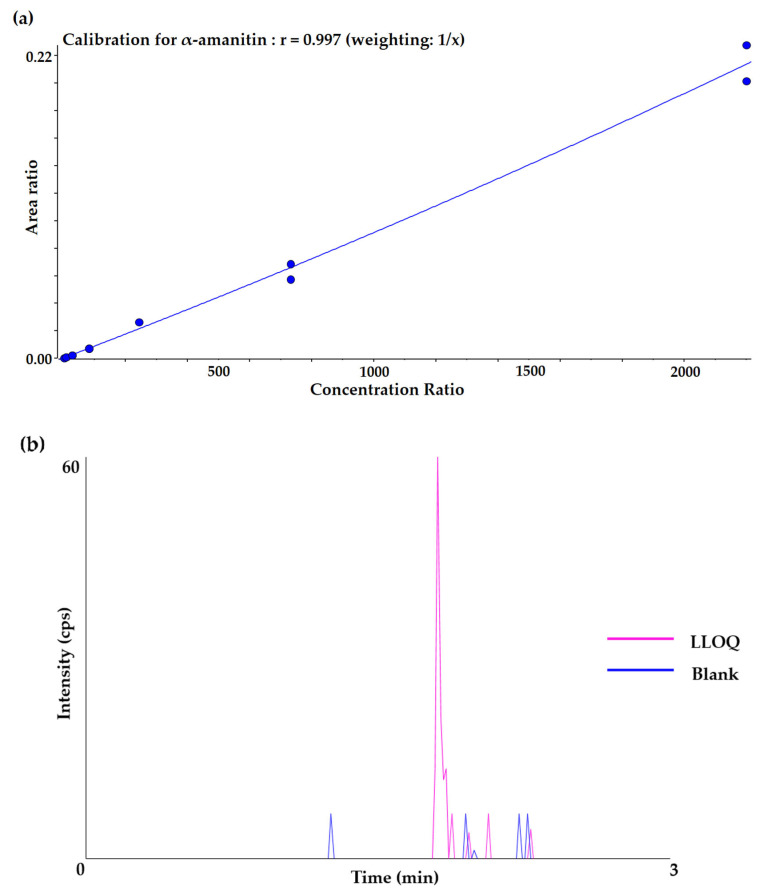
(**a**) Calibration curve for quantification and (**b**) chromatogram of LLOQ and blank samples of α-amanitin in rat plasma.

**Figure 3 molecules-27-08612-f003:**
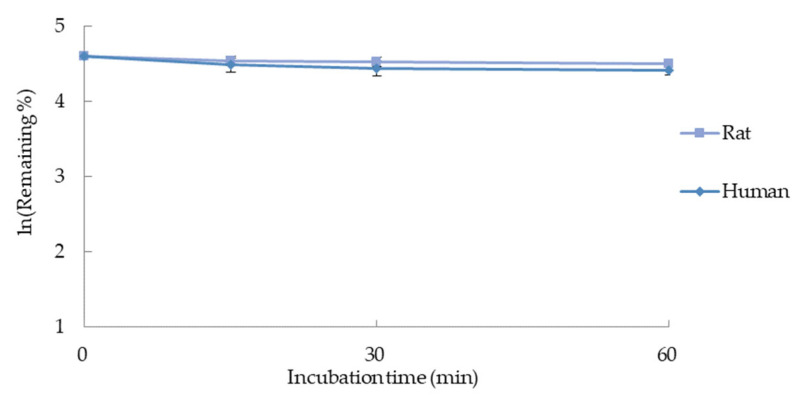
Metabolic stability of α-amanitin in rat and human liver microsomes.

**Figure 4 molecules-27-08612-f004:**
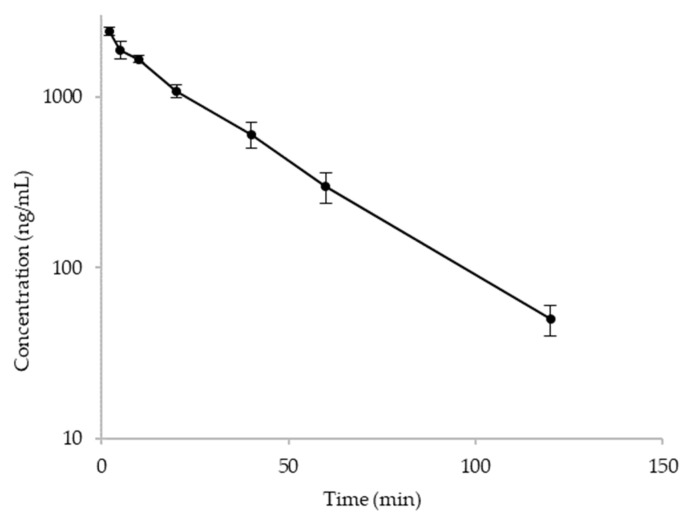
Time to concentration profile of α-amanitin in rats after IV administration at 1 mg/kg.

**Figure 5 molecules-27-08612-f005:**
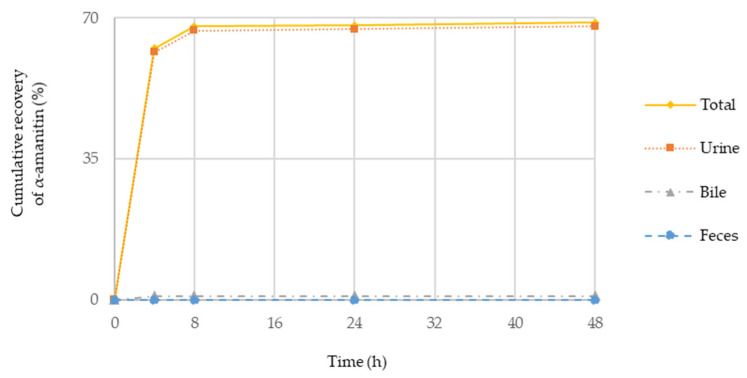
Cumulative recovery of α-amanitin in urine, bile, and feces after IV administration.

**Figure 6 molecules-27-08612-f006:**
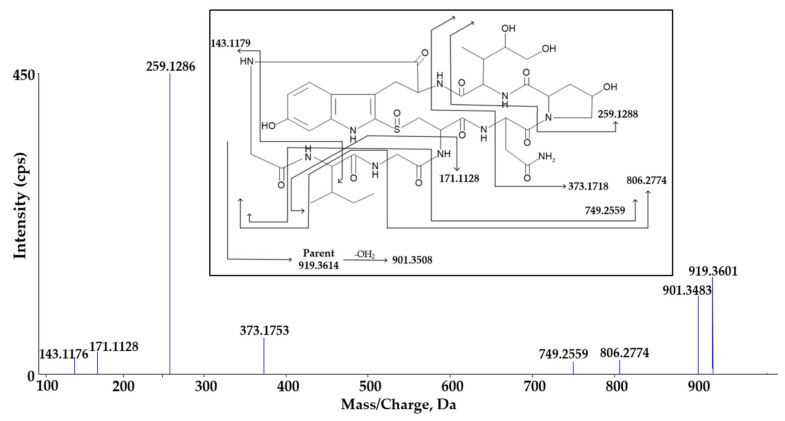
Fragmentation pattern of parent ion of α-amanitin (*m/z* 919.36).

**Figure 7 molecules-27-08612-f007:**
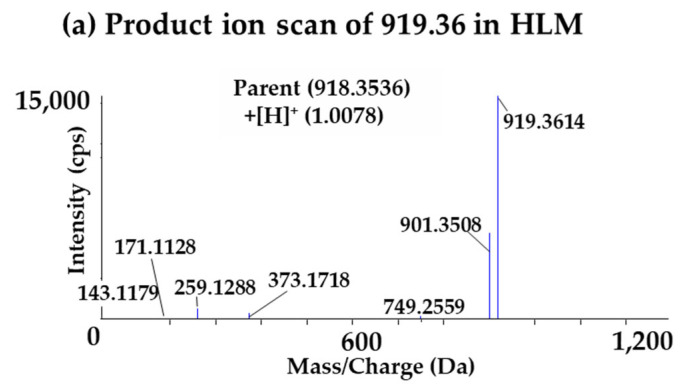
Identification of in vitro glucuronide metabolite of α-amanitin. (**a**) Product ion scan of α-amanitin in HLM (*m/z* 919.36); (**b**) product ion scan of α-amanitin in CLM (*m/z* 919.36); (**c**) product ion scan of a glucuronide metabolite in HLM (*m/z* 548.20); (**d**) product ion scan of a glucuronide metabolite in CLM (*m/z* 548.20).

**Figure 8 molecules-27-08612-f008:**
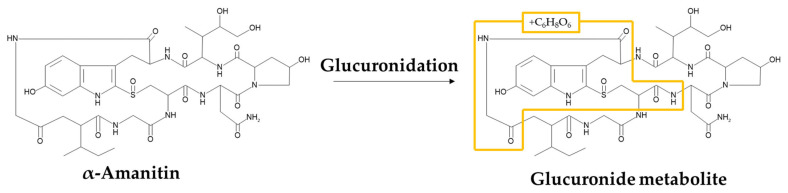
Metabolic pathways of α-amanitin under in vitro MetID (CLM and HLM).

**Figure 9 molecules-27-08612-f009:**
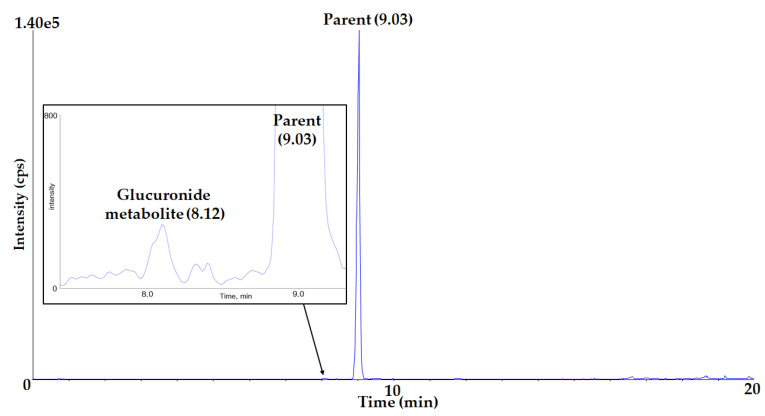
Total ion chromatograms of α-amanitin and its glucuronide metabolite in liver microsomes.

**Table 1 molecules-27-08612-t001:** QC results and statistics from the intra/inter-run assays for α-amanitin in rat plasma.

Run	Statistics	Low QC	Medium QC	High QC
		(15.0 ng/mL)	(165 ng/mL)	(1820 ng/mL)
Intra-run 1	Mean accuracy (%)	107	103	121
	Precision (% CV)	13.1	4.63	3.83
Intra-run 2	Mean accuracy (%)	105	105	95.2
	Precision (% CV)	6.45	5.15	3.72
Intra-run 3	Mean accuracy (%)	97.8	97.0	103
	Precision (% CV)	5.89	8.28	1.84
Inter-run	Mean accuracy (%)	103	102	107
	Precision (% CV)	8.55	6.02	10.5

**Table 2 molecules-27-08612-t002:** Stability tests for α-amanitin in rat plasma.

Stability	Statistics	Low QC	Medium QC	High QC
Assessment		(15.0 ng/mL)	(165 ng/mL)	(1820 ng/mL)
Short-term	Mean accuracy (%)	81.6	80.5	95.7
(4 h, RT)	Precision (% CV)	4.16	5.29	6.06
Long-term	Mean accuracy (%)	106	118	103
(2 weeks, –20 °C)	Precision (% CV)	5.31	4.23	7.89
Freeze-thaw	Mean accuracy (%)	94.9	103	114
(3 cycles, –20 °C and RT)	Precision (% CV)	3.10	6.60	2.80

**Table 3 molecules-27-08612-t003:** Extraction recovery of α-amanitin in rat plasma.

Run	The Extraction Recovery (%)
1	32.0
2	34.0
3	28.7
4	26.7
5	33.6
mean	31.0
% CV	10.2

**Table 4 molecules-27-08612-t004:** Remaining % of α-amanitin in rat and human liver microsomes.

Incubation Time (min)	Remaining %	
	Rat	Human
0	100	100
15	94.2 ± 8.60	89.2 ± 5.72
30	93.1 ± 8.64	85.1 ± 5.52
60	90.3 ± 4.37	82.6 ± 3.96

**Table 5 molecules-27-08612-t005:** Metabolic stability parameters of α-amanitin in rat and human liver microsomes.

Species	T_1/2_ (min)	CL_int_, In Vitro (mL/min/mg)	CL_int_, In Vivo (mL/min/kg)	CL_h_ (mL/min/kg)
Rat	447	<0.01	5.56	5.05
Human	236	<0.01	7.36	5.43

**Table 6 molecules-27-08612-t006:** PK parameters of α-amanitin in rats after IV administration at 1 mg/kg.

T_1/2_ (min)	C_max_ (ng/mL)	AUC_last_ (min·ng/mL)	Cl (mL/min/kg)	V_ss_ (mL/kg)
22.3 ± 0.98	2290 ± 274	70,400 ± 6030	14.0 ± 1.30	409 ± 18.3

**Table 7 molecules-27-08612-t007:** Recovery in urine, bile, and feces at IV and PO administration.

Route of Administration	Route of Excretion	Recovery (%)
IV	Urine	67.9
	Bile	1.00
	Feces	<0.01
PO	Urine	1.45
	Bile	0.05
	Feces	22.0

**Table 8 molecules-27-08612-t008:** Characterization of α-amanitin and its metabolite using the LC-qTOF-MS assay.

Metabolites	Observed *m/z*	Formula	Retention Time (min)	Error ppm	CLM and HLM	MLM, RLM, and DLM	Pooled Rat Plasma
Parent	919.3621	C_39_H_54_N_10_O_14_S	9.04	0.6	O	O	O
Glucuronide	548.2005	C_45_H_62_N_10_O_20_S	8.12	0.2	O	N/D *	N/D *

* N/D: Not detected.

**Table 9 molecules-27-08612-t009:** Mobile phase conditions for LC gradient.

**Quantification of α-Amanitin**	
**Time (min)**	**Mobile Phase B (%)**
0	5
0.6	5
1	95
1.3	95
1.4	5
3.0	5
**MetID for α-Amanitin**	
**Time (min)**	**Mobile Phase B (%)**
0	2
0.5	2
3.0	35
17.0	35
17.5	95
18.5	95
18.6	2
20.0	2

## Data Availability

The data presented in this study are available on request from the corresponding author.

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
