# Peer review of "Investigation of In Vitro and In Vivo Metabolism of α-Amanitin in Rats Using Liquid Chromatography-Quadrupole Time-of-Flight Mass Spectrometric Method"

_molecules, 2022, doi:10.3390/molecules27238612_

Round 1

Reviewer 1 Report

 This manuscript present a study on the disposition of alpha-amanitin in vivo in rat  and in vitro in rat and human hepatic microsomes and a few others.

The study is well written and clearly exposed.

Thus I have only a few remarks on the presentation and exploitation of the data.

Comments:

            Line 23 of the abstract suppress not in not detected.

            Page 2 line 46 : You could also give this reference on the crystal structure of the complex with polymerase II : doi:10.1073/pnas.251664698.

            Line 152 : you should describe what are solution A and B of Corning. Usually one state the concentration of each co-factor and co-enzyme ( thus G6P (mM), G6P-Dehydrogenase (nb unit/mL), NADPH, GSH, UDPGA (mM)

            Around line 193 : You use the m/z of the mono-charged and bi-charged ions. Have you tried to use also specific transition in CID mode to build a CID based method. (that could help people with a tripleQuad and no HRMS).

            Around line 200, concerning the recovery, you could perhaps try a solid phase extraction method. It may be more efficient than simple solvent precipitation of proteins. (or not)

         Page 9 : You have data from the in vivo showing production of a glucuronide. It looks that you used such conditions for making glucuronides in rat and human microsomes except you did not add alamethicin. Generally one add alamethicin to the microsomes to suppress the latency of the UDP- glucuronosyl transferases and add UDP-glucuronic acid as cosubstrate.

       Line 370 : What is the precision of this ion m/z (in ppm or mmu compared to the calculated exact mass?

       Figure 6 : Why did you not try a negative electrospray in order to test for the glucuronide. Often this is a more sensitive method for glucuronides than ESI+. For glucuronides one can also look for neutral loss of 176. For GSH conjugates monitoring the fragment at 272 in ESI negative  is also a good method.

       Paragraph bottom page 14 : It looks that the microsomes did modify part of the alpha-amanitin. Have you looked at possible metabolites (more that 10% transformation, using filtering of the mass data by diverse filters, like mass defect, or matrix subtraction? or difference between different time points. (Many paper using the Sciex 5600 and such techniques have been published (it is mainly post acquisition calculation using the right software. (see DOI: 10.1002/rcm.3585, DOI: 10.1021/ac071119u, 10.1021/tx1000046 ). Theoritically a High resolution method should allow such findings.

I know this is not your main purpose.

All together the paper presents interesting data and is well written.

There are a few remarks that should be answered . Thus minor revision.

Author Response

Thank you for reviewing.

Reviewer 2 Report

The main purpose of this manuscript is to explore the difference of in vitro-in vivo clearance and the possibility of extra-hepatic metabolism of α-amanitin. However, from the current experimental design and results, there is no definite conclusion.

1.     After α-amanitin was intravenously administrated to rats, it was mainly excreted in the form of parent drug via rat urine. After an oral administration, no metabolites were detected in rat plasma, urine and feces, suggesting whether administered intravenously or by gavage, α-amanitin was not eliminated from the body as the metabolites. Therefore, it was not necessary to study in vitro liver and extrahepatic metabolism.

2.     In Line 421, the authors mentioned that the bioavailability of α-amanitin in rats was almost 0%, but from the data of urinary excretion (Table 8), the bioavailability of α-amanitin was more than 2%. The results of the excretion experiment also showed that the cause of the low bioavailability of α-amanitin was related to its poor oral absorption, rather than to its extrahepatic metabolism, as the authors postulated.

3.     Based on the data in Table 5, the metabolic stability parameters as shown in Table 6 could not be obtained and the authors should check the relevant data.

4.     In the study, an LC-qTOF-MS method was developed to detemine α-amanitin in rat plasma, urine and feces samples. However, compared with the methods reported in the literature (J Toxicol Environ Health, 2021, 84: 821-835), the accuracy and sensitivity were not satisfactory. This may also be the reason why α-amanitin in rat plasam after intragastric administration were not detected in this manuscript.

5.     In addition, after intragastric administration, the recovery of α-amanitin was significantly lower than that of the intravenous administration group, which may be related to the strong binding to the surface of the gastrointestinal tract, the authors explained. In fact, this manuscript did not provide the extraction recovery of sample preparation for the determination of α-amanitin in rat feces. The recovery of α-amanitin in plasma was only 31%. It was also possible that α-amanitin in fecal samples has not been effectively extracted, resulting in low concentrations measured in feces.

Author Response

Thank you for reviewing.
